# Polyaromatic Carboxylate Ligands Based Zn(II) Coordination Polymers for Ultrasound-Assisted One-Pot Tandem Deacetalization–Knoevenagel Reactions

Anirban Karmakar [1,*], Mohamed M. A. Soliman [1,2], Elisabete C. B. A. Alegria [1,2], Maria Fátima C. Guedes da Silva [1,3] and Armando J. L. Pombeiro [1,4,*]

1 Centro de Química Estrutural, Institute of Molecular Sciences, Instituto Superior Técnico, Universidade de Lisboa, Av. Rovisco Pais, 1049-001 Lisboa, Portugal; mohamed.soliman@tecnico.ulisboa.pt (M.M.A.S.); elisabete.alegria@isel.pt (E.C.B.A.A.); fatima.guedes@tecnico.ulisboa.pt (M.F.C.G.d.S.)
2 Chemical Engineering Departament, Instituto Superior de Engenharia de Lisboa, Instituto Politécnico de Lisboa, R. Conselheiro Emídio Navarro, 1, 1959-007 Lisboa, Portugal
3 Departamento de Engenharia Química, Instituto Superior Técnico, Universidade de Lisboa, Av. Rovisco Pais, 1049-001 Lisboa, Portugal
4 Peoples' Friendship University of Russia (RUDN University), Research Institute of Chemistry, 6 Miklukho-Maklaya Street, 117198 Moscow, Russia
* Correspondence: anirban.karmakar@tecnico.ulisboa.pt (A.K.); pombeiro@tecnico.ulisboa.pt (A.J.L.P.)

**Abstract:** Solvothermal reactions between the polyaromatic group containing carboxylic acid pro-ligands 5-{(pyren-1-ylmethyl)amino}isophthalic acid (**H2L1**) and 5-{(anthracen-9-ylmethyl)amino} isophthalic acid (**H2L2**) with $Zn(NO_3)_2 \cdot 6H_2O$ led to the formation of the new 1D coordination polymer $[Zn(L1)(NMF)]_n$ (**1**) and four other coordination polymers, $[Zn(L1)(DMF)]_n$ (**2**), $[Zn(L1)(4,4'\text{-}Bipy)]_n$ (**3**), $[Zn(L2)(DMF)(H_2O)_2]_n \cdot n(H_2O)$ (**4**) and $[Zn_2(L2)_2(DMF)(CH_3OH)]_n$ (**5**), which were previously reported by our group. Single crystal X-ray diffraction analyses revealed that the CP **1** has a one-dimensional (1D) double-chain-type structure similar to that of CP **2**. For CP **3**, the assembly of the Zn(II) ion with a deprotonated $L1^{2-}$ ligand and 4,4'-bipyridine produces a 3D network. CP **4** and **5** exhibit 1D linear and 2D layered-type structures. The ultrasound-assisted tandem reactions promoted by CPs have not yet been well studied. Thus, in the present work, we have investigated the catalytic activities of the newly synthesized CP **1**, as well as of the other CPs **2–5**, towards the tandem deacetalization–Knoevenagel condensation reactions of various acetals under ultrasonic irradiation. They proved to be highly efficient, with special emphasis on catalyst **1**, which completely converted the substrate (benzaldehyde dimethyl acetal) into the desired product (2-benzylidenemalononitrile) after 2 h. The stability of the catalysts, namely regarding the action of ultrasonic radiation, was demonstrated by their reuse, where only a slight loss of activity was observed after four cycles. Heterogeneity was also demonstrated, and no leaching was detected over the various cycles.

**Keywords:** coordination polymer; crystal structure analysis; heterogeneous catalysis; tandem reactions; deacetalization–Knoevenagel condensation reaction

## 1. Introduction

Coordination polymers (CPs) are a class of porous crystalline materials consisting of metallic centers linked together by multidentate organic ligands comprising ordered networks with different dimensions [1,2]. The large surface area and porosity, as well as the structural flexibility, are unique features that provide enormous visibility and usefulness to these structures. As a result, a generation of functional CPs emerged and have been presented as excellent candidates for diverse applications, namely as gas storage materials, sensing, photocatalysts, heterogenous catalysts and drug delivery materials [3–5].

The catalytic performance of a CP is influenced by the active sites generated by the chosen metallic centers and the bridging ligands [6]. In addition, a second category of active

sites can be added, which can significantly modify the catalytic activity of a CP. The generation of an active site can be guaranteed in different ways: by removing solvent molecules from the CP structure, by creating defects in their structure, by introducing functional organic sites during their construction, or by incorporating catalytic active species in their cavities [7–11]. Thus, recently our group has prepared various functionalized carboxylate ligands by introducing amine (-NH) and amide (-NHCO) groups and synthesized several amine/amide functionalized CPs, where the amine/amide group acts as a Lewis basic center and the metal as a Lewis acid, and these CPs show promising catalytic behaviors for various organic transformations [12].

On the other hand, the employment of multifunctional MOFs/CPs in tandem reactions, which are distinct reactions performed in one-pot (cascade reactions), have been reported [13,14]. Normally, the reactants are activated by one active site, whereas a formed intermediate is activated by the second type of active site. In this sense, and considering that the intermediate is not isolated, tandem reactions reduce energy consumption and waste by decreasing the number of reaction steps and minimizing the use of solvents and reagents [15]. Hence, they follow the principles of green chemistry.

Zhou's group designed and constructed a porous Cu(II)-based coordination network (PCN-124) where the collective action of the Cu(II) Lewis acid centers and Lewis basic pyridine and amide groups from the 5,5′-((pyridine-3,5-dicarbonyl)bis(azanediyl)) diisophthalate ligand makes such a network an efficient catalyst for one-pot deacetalization–Knoevenagel tandem reactions [16]. The MIL-101(Al)-NH$_2$ ([Al(OH)(NH$_2$-BDC)]$_n$, where BDC = 1,4-benzenedicarboxylate) MOF, due to the presence of the Brønsted acidic carboxylic acid (COOH) moieties and the unsaturated Al(III) Lewis acid sites as well as amine basic sites, acts as a promising bifunctional acid–base catalyst for the one-pot tandem deacetalization–Knoevenagel condensation [17]. However, many MOF/CP-based catalysts require a high catalyst loading, long reaction time and high reaction temperature [18]. Thus, the synthesis of heterogeneous catalysts that can catalyze multi-step tandem reactions under mild conditions is worth developing. In this context, we have already explored the catalytic activity of different Zn(II) and Cd(II)-based amido carboxylate CPs toward the one-pot deacetalization–Knoevenagel tandem reaction and achieved a high catalytic activity [19,20]. However, the catalytic activities of CPs based on polyaromatic and amine groups containing carboxylate ligands were not explored.

In the last decade, the use of ultrasound irradiation to promote catalyst-free organic reactions has significantly emerged [21]. Ultrasound irradiated reactions can be more beneficial than those undertaken under traditional thermal methods in the context of reaction yields, time, purity and selectivity of the products, etc. [21]. However, examples of ultrasound-assisted cascade reactions are rare [20]. Thus, exploring suitable CP-based catalysts for ultrasonic-assisted tandem reactions is a topic of great interest.

Hence, in the present work, two polyaromatic and amine groups containing carboxylic acid-based pro-ligands, namely 5-{(pyren-1-ylmethyl)amino}isophthalic acid (**H$_2$L1**) and 5-{(anthracene-9-ylmethyl)amino}isophthalic acid (**H$_2$L2**) [22], are employed as linkers, under solvothermal conditions, for the synthesis of a new 1D coordination polymer [Zn(L1)(NMF)]$_n$ (**1**) and four other known CPs, [Zn(L1)(DMF)]$_n$ (**2**), [Zn(L1)(4,4′-Bipy)]$_n$ (**3**), [Zn(L2)(DMF)(H$_2$O)$_2$]$_n$·n(H$_2$O) (**4**) and [Zn$_2$(L2)$_2$(DMF)(CH$_3$OH)]$_n$ (**5**), which were previously reported by our group. The synthesis of **H$_2$L1** and **H$_2$L2** as well as the synthesis, structure and toxic organic dye removal properties of **2–5** were already reported by our group [22], but the effectiveness of these CPs as heterogeneous catalysts in organic reactions has not been studied. Thus, on account of the above considerations, we have tested the catalytic activities of **1–5** towards the one-pot tandem deacetalization–Knoevenagel condensation reaction with different acetals and malononitrile under ultrasonic irradiation. Moreover, we have also studied the recyclability and confirmed the heterogeneity of these catalysts.

## 2. Results and Discussion

### 2.1. Characterization of CP **1**

In the FT-IR spectrum, the strong bands for the carboxylate ($COO^-$) groups appear at 1681 cm$^{-1}$ for asymmetric stretching and 1337 cm$^{-1}$ for symmetric stretching. The NH-stretching vibrations of the amine groups are detected at 3387 cm$^{-1}$.

Thermogravimetric analysis of CP **1** was carried out in the range from 30–800 °C at a heating rate of 5 °C min$^{-1}$ under nitrogen. As shown in Figure S1 (Supporting Information), CP **1** shows a weight loss of 11.8% between 78 °C and 225 °C, corresponding to the loss of one coordinated N-methylformamide molecule (calcd: 11.4%) from the asymmetric unit. Upon further heating, the framework starts to decompose slowly until 297 °C, and after 298 °C, it starts to decompose more rapidly until 800 °C.

The $N_2$ and $CO_2$ gas adsorption studies of CP **1** were performed after the activation of the sample under vacuum at 110 °C for 8 h (Figure 1). The $N_2$ adsorption study was performed at 77 K, and the adsorption–desorption shows a reversible type II isotherm. The Brunauer–Emmett–Teller (BET) surface area of CP **1** was ca. 39.0 m$^2$g$^{-1}$, which is comparable to the BET surface area of CP **2** (36.4 m$^2$g$^{-1}$) [22]. Recently, various amine-functionalized CPs show highly efficient carbon dioxide adsorption properties [20]. Thus, we have also analyzed the $CO_2$ adsorption properties of CP **1,** and the maximum $CO_2$ adsorption reached 20.2 cm$^3$ g$^{-1}$ at 273 K and 900 mmHg (Figure 1B).

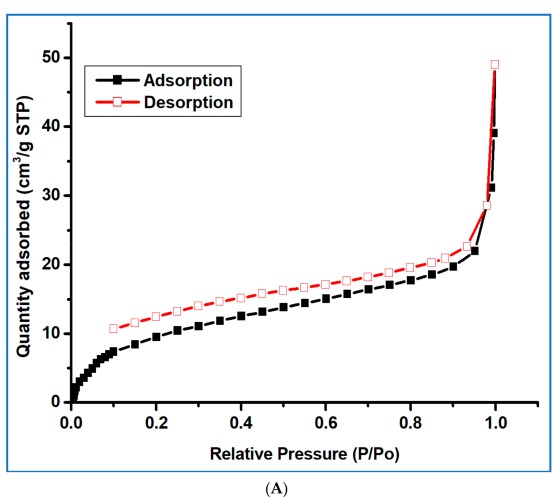

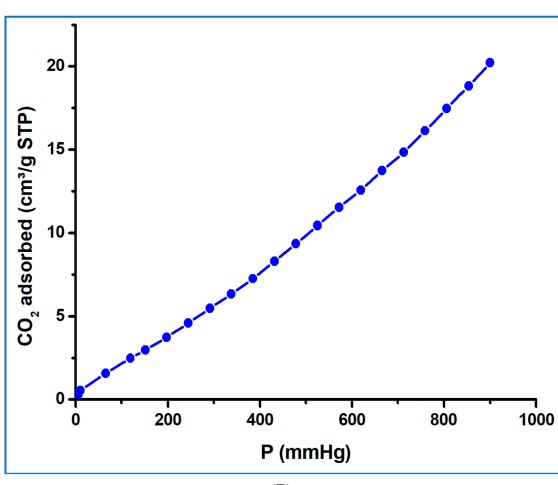

(A)

(B)

**Figure 1.** (**A**) $N_2$ sorption isotherm of framework **1** at 77 K. (**B**) $CO_2$ sorption isotherm of **1** at 273 K.

### 2.2. Crystal Structure Analysis

The crystal structures of CP **2–5** were previously reported by us [22], but their description is included now in a comparative way. The crystal X-ray diffraction analysis revealed that CP **1** has a one-dimensional structure similar to that of CP **2**; they are pseudo-polymorphic one-dimensional isomers. Their asymmetric units consist of one Zn(II) ion, one L1$^{2-}$ ligand and one coordinated N-methylformamide (NMF) (for CP **1**) (Figure 2A) or N,N′-dimethylformamide (DMF) (for CP **2**) molecule. In both frameworks, the zinc(II) metal centers own distorted square pyramidal geometries ($\tau_5 = 0.27$–0.28) [23], defined by four carboxylate-O from three L1$^{2-}$ ligands in the equatorial positions and the axial site being coordinated by the oxygen from an NMF (for CP **1**) or a DMF molecule (for CP **2**). In these CPs, the coordinated L1$^{2-}$ ligands are twisted, and the dihedral angle between the pyrene and the aminoisophthalate unit is in the range of 79.05° to 79.17°. In both frameworks, the carboxylate groups of the L1$^{2-}$ ligand bind the Zn(II) centers in a 1$\kappa O$,2$\kappa O'$:3$\kappa O''O''''$ bridge bidentate and chelating fashion and generate a binuclear [Zn$_2$(COO)$_4$] secondary building unit (SBU). These are assembled with the deprotonated L1$^{2-}$ ligands to produce a similar one-dimensional double chain-type structure (Figures 2C and S2). In CP **1**, the distance between two symmetry-related Zn(II) ions is similar to that in CP **2** (3.932 Å

for CP **1** and 3.927 Å for CP **2**). The one-dimensional chains are interconnected through N-H⋯π interactions (between amine NH and pyrene rings, $d_{D-A}$ 3.420–3.424 Å range) and via C-H⋯π interactions (between methylene $CH_2$ and pyrene ring, $d_{D-\pi}$ 3.609–3.620 Å) (Figure 2D) and produce a 2D hydrogen-bonded network. Besides that, several C-H⋯O interactions between the pyrene-CH and the carboxylate-O atoms help these frameworks to expand into the third dimension.

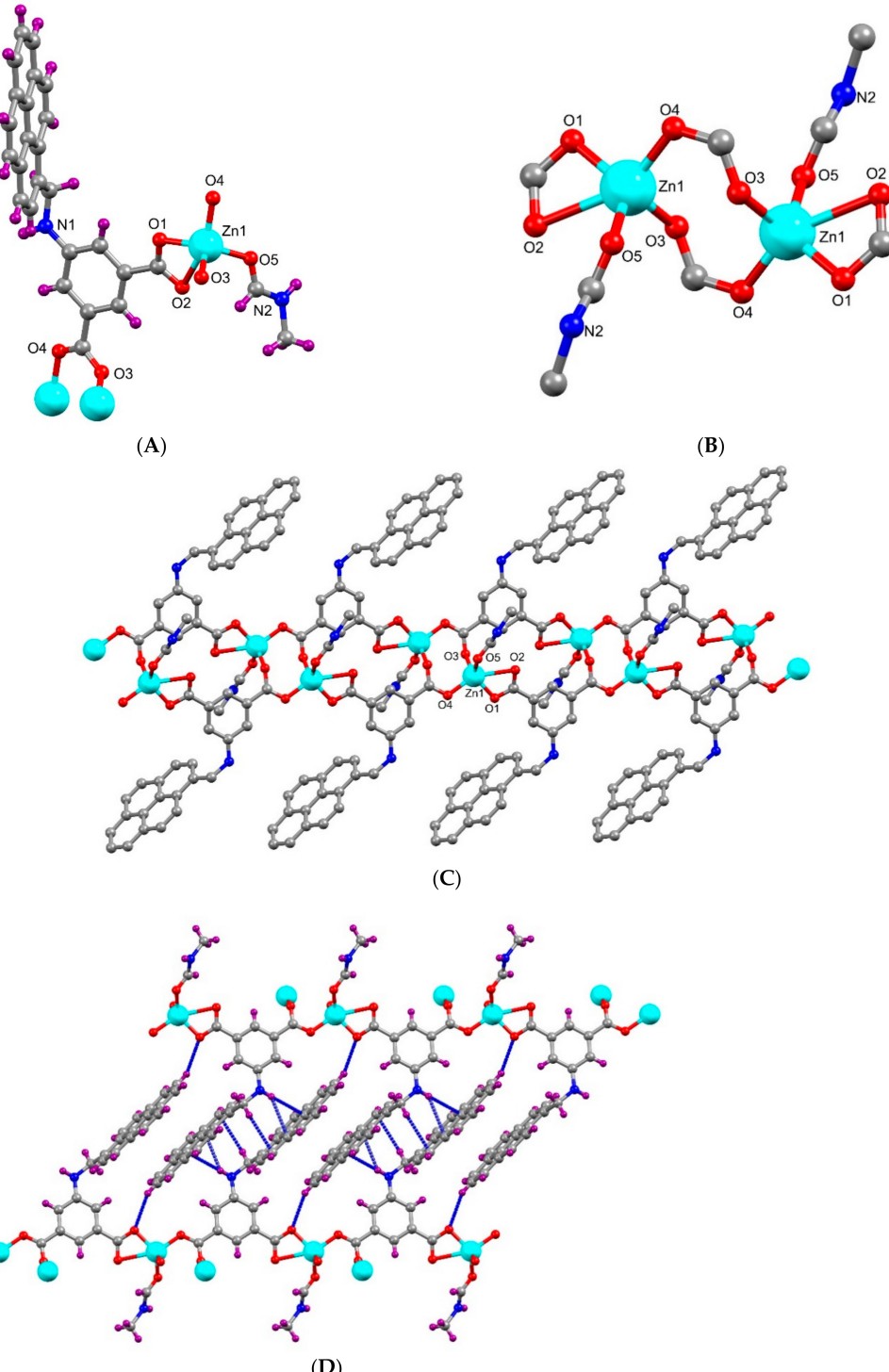

(A)

(B)

(C)

(D)

**Figure 2.** (**A**) Molecular structure of CP **1** with partial atom labelling scheme. (**B**) Schematic representation of binuclear SBU. (**C**) One-dimensional double chain-type structure of CP **1**. (**D**) Hydrogen bonding between two 1D chains of CP **1**.

Like in CP **1**, the binuclear [Zn$_2$(COO)$_4$] unit in CP **3** acts as an SBU, which further connects with L1$^{2-}$ and 4,4′-bipyridine ligands to generate a three-dimensional framework (Figure S2B). In CP **4**, the Zn(II) center presents a distorted octahedral geometry, and no SBU was observed; the combination of L2$^{2-}$ ligand with Zn(II) ions led to the development of a one-dimensional coordination polymer (Figure S3A). In CP **5,** the assembly of binuclear [Zn$_2$(COO)$_4$(DMF)(MeOH)] unit with L2$^{2-}$ ligands produced a two-dimensional network (Figure S3B).

*2.3. Ultrasound-Assisted One-Pot Deacetalization–Knoevenagel Tandem Reactions*

The Zn(II) polymeric compounds **1–5** were tested as heterogenous catalysts for the synthesis of α,β-unsaturated dinitriles through ultrasound-assisted one-pot deacetalization–Knoevenagel tandem reaction. The selected reaction involves two sequential steps: first, the in situ formation of the intermediate benzaldehyde (Scheme 1 A→B), through the deacetalization of benzaldehyde dimethylacetal (A) catalyzed by the Zn(II) centers; second, the generation of 2-benzylidenemalononitrile (Scheme 1, B→D) by Knoevenagel condensation, which is expected [20,24] to involve the carbonyl group of benzaldehyde (Scheme 1, B) and the cyano group of malononitrile (Scheme 1, C) interactions with the Zn(II) centers, generating 2-benzylidenemalononitrile (Scheme 1, D). The acidity of the methylene group of malononitrile facilitates the abstraction of protons by basic centers, generating a nucleophilic species that further attacks the carbonyl group of benzaldehyde to form a C-C bond, followed by dehydration to result in the formation of 2-benzylidenomalononitrile.

(*in situ*)

**Scheme 1.** One-pot cascade deacetalization–Knoevenagel condensation reaction catalyzed by **1–5** Zn(II) coordination polymers (**A**—benzaldehyde dimethylacetal; **B**—benzaldehyde; **C**—malononitrile; **D**—2-benzylidenemalononitrile).

The results showed that the one-pot tandem reaction could be efficiently catalyzed by **1–5**, with almost complete conversion of benzaldehyde dimethyl acetal into 2-benzylidenemalononitrile (yield 99%) upon using 1 mol% of CP **1** as a catalyst, within 2 h of the reaction at 80 °C (Table 1, entry 1, Figure 3). By using CP **2** and **3**, yields of 96 and 97% were achieved within the same time and temperature, respectively (Table 1, entries 2 and 3). However, the anthracene ligand (**H$_2$L2**)-based CPs **4** and **5** show comparatively lower yields of 95 and 90%, respectively. Increasing the reaction time to 3 h did not significantly improve the yield for these CPs (Figure 3).

The lower product yields (compared to CP **1**) for CPs **3** and **5** may be due to their tightly packed non-porous 3D and 2D networks, which lower the accessibility of the metal centers. In the case of CP **4**, the strong hydrogen-bonded 3D network between the coordinated or non-coordinated water molecules and CP may also hamper the availability of the metal centers. Although CPs **1** and **2** have similar 1D weak hydrogen-bonded networks, the lower final product yield for the latter may be due to slightly stronger coordination of the DMF (Zn-O$_{DMF}$ 1.993 Å, for CP **2**) in comparison to the NMF (Zn-O$_{NMF}$ 2.001 Å, for

CP **1**), corresponding to lower lability of the DMF ligand, hampering the formation of open metal sites (OMS). Thus, the highest catalytic activity of CP **1** compared to other CPs is believed to result from its low-dimensional weak hydrogen-bonded structure, associated with the presence of a labile ligand (a coordinated NMF solvent molecule), which promote the accessibility and unsaturation of the metal centers.

**Table 1.** Deacetalization–Knoevenagel tandem reactions between benzaldehyde and malononitrile with CPs **1–5** as catalysts and under ultrasound irradiation [a].

| Entry | Catalyst | Time (h) | Catalyst (mol%) | T (°C) | Solvent | Unreacted A (%) [b] | Yield of B (%) [b] | Yield of D (%) [b] |
|---|---|---|---|---|---|---|---|---|
| | | | | Different Catalysts | | | | |
| 1 | **1** | 2 | 1 | 80 | DMF | 0 | 0 | >99 |
| 2 | **2** | 2 | 1 | 80 | DMF | 0 | 4 | 96 |
| 3 | **3** | 2 | 1 | 80 | DMF | 1 | 2 | 97 |
| 4 | **4** | 2 | 1 | 80 | DMF | 4 | 1 | 95 |
| 5 | **5** | 2 | 1 | 80 | DMF | 8 | 2 | 90 |
| | | | | Different Substrates | | | | |
| 6 [c] | **1** | 2 | 1 | 80 | DMF | 2 | 0 | 98 |
| 7 [d] | **1** | 2 | 1 | 80 | DMF | 4 | 0 | 96 |
| 8 [e] | **1** | 2 | 1 | 80 | DMF | 8 | 1 | 91 |
| 9 [f] | **1** | 2 | 1 | 80 | DMF | 14 | 8 | 78 |

[a] Typical reaction conditions: 1 mol% catalyst (1 mmol of monomeric unit of **1** per 100 mmol of substrate), solvent (0.5 mL), benzaldehyde dimethyl acetal (152 mg, 1.0 mmol) and malononitrile (132 mg, 2.0 mmol); 0.25–2 h, r.t., 50 or 80 °C, ultrasound irradiation (50/60 Hz for 80 °C). [b] Calculated by $^1$H NMR analysis (**A**—benzaldehyde dimethyl acetal; **B**—benzaldehyde; **D**—2-benzylidenemalononitrile). [c] 4-Bromobenzaldehyde dimethyl acetal used as substrate. [d] 4-Chlorobenzaldehyde dimethyl acetal used as substrate. [e] 4-Methoxybenzaldehyde dimethyl acetal used as substrate. [f] 3-bromo benzaldehyde diethyl acetal used as substrate.

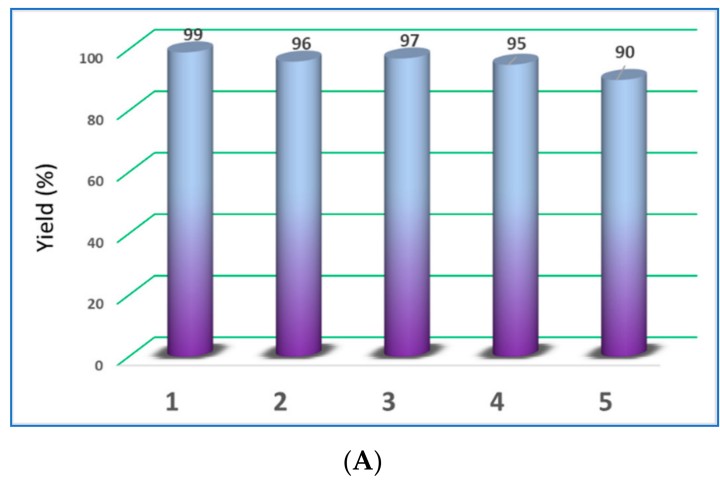
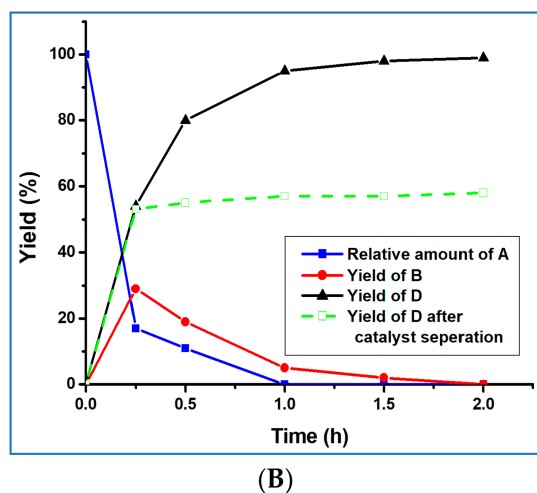

(**A**)　　　　　　　　　　　　　　　　　　　　　　(**B**)

**Figure 3.** (**A**) Yields of 2-benzylidenemalononitrile by using CPs **1–5** as catalysts. (**B**) Yield vs. time plot for the deacetalization–Knoevenagel condensation reactions catalyzed by CP **1** (black line: yield of 2-benzylidenemalononitrile (**D**); red line: yield of benzaldehyde (**B**); blue line: percentage of unreacted benzaldehyde dimethyl acetal (**A**); dotted green line: yield of 2-benzylidenemalononitrile (**D**) upon separating the catalyst after 0.25 h of reaction time).

The final product was not detected when no catalyst was added into the system (Table S3, entry 14). The zinc salt used in the CP preparations, $Zn(NO_3)_2 \cdot 6H_2O$, could efficiently catalyze the first reaction (deacetalization) to obtain benzaldehyde (82% yield), but no further condensation product was detected (Table S3, entry 15). Ligands $H_2L1$ and $H_2L2$ were also tested, and in this case, only the first step of the tandem reaction was

achieved, with 48 and 45% of benzaldehyde yield, respectively (Table S3, entries 16 and 17), without any conversion of benzaldehyde to 2-benzylidenemalononitrile.

Apart from DMF, different solvents, such as CH$_3$CN, EtOH and MeOH, were tested (Table S3, entries 3–5 for catalyst **1**) to inquire about the most suitable one. The tests were carried out with all catalysts (**1–5**) at 80 °C and for 2 h, and the results showed that DMF is the most promising solvent resulting in a 99% yield of 2-benzylidenemalononitrile. In the case of EtOH, MeOH and CH$_3$CN, although the conversions are high (above 80%), the yield of the final product drops to 63, 71 and 10% with EtOH, MeOH and CH$_3$CN, respectively. The reaction was also performed without any solvent (under added solvent-free conditions), and the yield of 2-benzylidenemalononitrile was ca. 85% (Table S3, entry 6).

Besides the standard catalyst load (1 mol%), a half amount was also tested for catalyst **1**, and the final product yield decreased from 99 to 63% (Table S3, entry 1). Additionally, when the amount of catalyst **1** was doubled (2 mol%), a complete conversion to 2-benzylidenemalononitrile was achieved (Table S3, entry 2). Since 1 mol% of catalyst led to almost a quantitative conversion, the remaining studies were performed with such a catalyst loading.

The temperature effect was also studied, and besides the typical 80 °C, the reactions were also undertaken at lower temperatures, namely 50 °C and room temperature. At room temperature, benzaldehyde dimethyl acetal (**A**) is completely converted into benzaldehyde (**B**), without any final 2-benzylidenemalononitrile (**D**) product, whereas at 50 °C, it is possible to observe the formation of **D** but in a yield that does not exceed 69% (Table S3, entries 7 and 8).

We also tried to optimize the reaction in terms of time by shortening it to shorter periods, namely 0.25, 0.5, 1 and 1.5 h. The kinetic plot of 2-benzylidenemalononitrile (**D**) formation vs. time is depicted in Figure 3B for catalyst **1**. The yield of the final product **D** is always increasing with a more accentuated growth in the first hour of reaction, having already reached 95% at the end of this period and 50% after 0.25 h of reaction. Furthermore, a test was carried out in which, after 0.25 h, the catalyst was removed from the reaction mixture, and the reaction was allowed to extend up to 2 h but now in the absence of a catalyst. As shown in Figure 3B (green dotted line), the formation of the final product was literally stopped after the catalyst suppression, proving the heterogeneity of the catalyst [25]. This is also confirmed by the negligible amount (0.023% of that used in the reaction) of Zn(II) ions in the solution after removing the catalyst from the reaction mixture, which also indicates that no leaching occurs during the catalytic process.

We have also tested the activity of our catalyst **1** towards a variety of substituted benzaldehyde dimethyl and diethyl acetals, and the obtained yields of the corresponding benzylidenemalononitriles range from 78 to 98% (entries 22–25, Table 1). The reaction yields using the *para*-substituted bromo (−Br) and chloro (−Cl) dimethyl acetals are comparable and in the range of 96–98% after 2 h of reaction time. However, for 4-methoxybenzaldehyde dimethyl acetal and 3-bromo benzaldehyde diethyl acetal, only 91% and 78% of the product yields were obtained, respectively.

The reusability of the catalyst is a relevant feature for the sustainability of a catalytic process. In this sense, our catalyst **1** was isolated after the first catalytic cycle, recovered and reused in six subsequent cycles (Figure 4). The catalyst activity was practically not affected in the first two cycles, having only decreased to a limited extent after the third and four cycles, and a further decrease occurred after the sixth cycle. We have also observed a similar phenomenon for the other 1D CPs **2** and **4** (Figure S5). However, for the 3D CP **3** and 2D CP **5**, a slight decrease in yields was observed only after the fourth cycle (Figure S5). The structural integrity of **1–5** during the catalytic process was confirmed by FT-IR and powder X-ray diffraction analyses performed before and after its use (Figures S3 and S5). The similarity between the FT-IR spectra and the XRD powder diffractograms before and after catalytic reaction supports that the structure of the catalyst remains intact after the catalytic reaction.

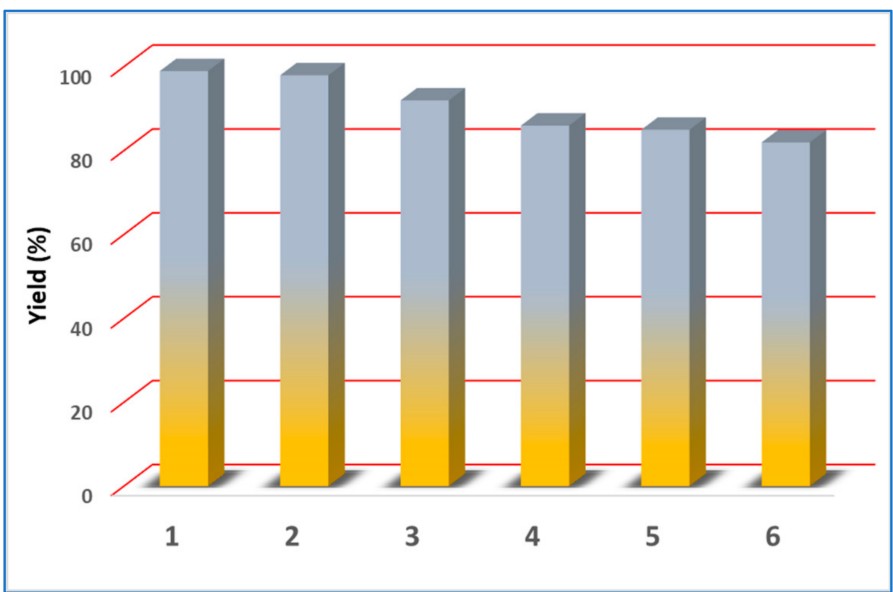

**Figure 4.** Yield of 2-benzylidenemalononitrile (**D**) upon catalyst recycling for the deacetalization–Knoevenagel reactions catalyzed by CP **1**.

Although the ultrasound-assisted route for the one-pot deacetalization–Knoevenagel condensation is not yet widely explored, a comparison of the activity of our catalysts **1**–**5** with other reported MOFs under this technique demonstrates that our Zn(II) compounds **1**–**5** (leading to product yields of 90–99%, within 2 h and with 1 mol% catalyst) are competing for catalysts that should not be underestimated.

For example, these results are comparable to those reported by some of us [20] with the binuclear metallomacrocyclic Zn(II) compounds $[Zn_2(L)_2(H_2O)_4]\cdot 2(H_2O)\cdot 6(DMF)$ and $[Zn_5(L)_4(OH)_2(H_2O)_4]n\cdot 8n(DMF)\cdot 4n(H_2O)$ [$H_2L$ = 4,4′-{(pyridine-2,6-dicarbonyl)bis(azanediyl)}dibenzoic acid], whose catalytic activity has been markedly improved with the application of ultrasonic irradiation (in comparison to microwave irradiation and conventional heating), leading to 98 and 99% of 2-benzylidenemalononitrile, respectively, within 2 h at 80 °C in DMF (0.5 mL), under ultrasound irradiation. On the other hand, silica sodium carbonate nanoparticles [26] have been used for the preparation of arylidene malononitriles/methylciano or ethylciano acetates and achieved 88% of product yield within 2 h with 1 mol% catalyst in $CH_3CN$ (10 mL) when sonicated at 70 °C.

Other Zn(II)-based MOFs have been applied for the deacetalization–Knoevenagel reaction under conventional heating, e.g., the hybrid material $\{(Me_2NH_2)[InZn(TDP)(OH_2)]$ $\cdot 4DMF\cdot 4H_2O\}_n$, resulting from the combination of In(III) and Zn(II) in the presence of a structure-oriented $TDP^{6-}$ [2,4,6-tri(2,4-dicarboxyphenyl)pyridine] ligand, which leads to complete conversion after 6 h of reaction time (longer than ours) under conventional heating (60 °C) [27]. Moreover, the 2D Zn(II) MOF $[Zn_2(L')(H_2O)4]n\cdot 4n(H_2O)$, linked with 5,5′-{(pyridine-2,6-dicarbonyl)bis(azanediyl)}diisophthalate (L′), also leads to full conversion after 3 h (longer time than in our case) at 75 °C [19]. For MIL-101(Al)-$NH_2$, a yield of 94% is obtained at 90 °C after 3 h of reaction time [17]. With another polymeric Zn(II) framework, $[Zn_4(TBCB)(H_2O)_6]_n\cdot 5n(DMAc)$ (where TBCB = 2,2′,6,6′-tetrakis[3,5-bis-3,5-benzenedicarboxylate]benzidine, DMAc = N,N-dimethylacetamide), a 99% yield is achieved after 4 h at 90 °C, which is a higher reaction time and temperature than ours [28]. With our catalyst **1**, we have obtained >99% of yield after 2 h at 80 °C under ultrasonic irradiation.

## 3. Materials and Methods

The synthetic work was performed in air and at a temperature that was higher than the room temperature. All the chemicals were obtained from commercial sources and used

as received. The infrared spectra (4000–400 cm$^{-1}$) were recorded on a Bruker Vertex 70 instrument in KBr pellets; abbreviations: s = strong, m = medium, w = weak, bs = broad and strong, mb = medium and broad. The $^1$H NMR spectra were recorded at ambient temperature on a Bruker Avance II + 300 (UltraShield$^{TM}$Magnet) spectrometer operating at 300.130 MHz. The chemical shifts are reported in ppm using tetramethylsilane as the internal reference (abbreviations: s = singlet, d = doublet, t = triplet, q = quartet). Carbon, hydrogen and nitrogen elemental analyses were carried out by the Microanalytical Service of the Instituto Superior Técnico using a PerkinElmer Series II 2400 and a SPECTRO Inductively Coupled Plasma Atomic Emission Spectroscopy (ICP-AES) instrument, respectively. X-ray quality single crystals of the CP **1** were immersed in cryo-oil, mounted in a nylon loop and measured at room temperature. Intensity data were collected using a Bruker APEX-II diffractometer. Thermal properties were analyzed with a Perkin-Elmer Instrument system (STA6000) at a heating rate of 5 °C min$^{-1}$ under a dinitrogen atmosphere. Powder X-ray diffraction (PXRD) was conducted in a D8 Advance Bruker AXS (Bragg Brentano geometry) theta-2-theta diffractometer, with copper radiation (Cu K$\alpha$, $\lambda$ = 1.5406 Å) and a secondary monochromator, operated at 40 kV and 40 mA. Flat plate configuration was used, and the typical data collection range was between 5° and 40°. N$_2$ and CO$_2$ adsorption analysis of activated CP **1** (under vacuum at 100 °C for 10 h) was carried out at 77 K and 273 K with a Micromeritics ASAP 2060 instrument.

### 3.1. Synthesis of the Pro-Ligands $H_2L1$ and $H_2L2$

The polyaromatic group functionalized pro-ligands, 5-{(pyren-1-ylmethyl)amino} isophthalic acid ($H_2L1$) and 5-{(anthracen-9-ylmethyl)amino}isophthalic acid ($H_2L2$), were synthesized and characterized as already reported by us [22].

### 3.2. Synthesis of Coordination Polymer 1

The solvothermal reaction of $H_2L1$ with Zn(NO$_3$)$_2$·6H$_2$O in a mixture of N-methylformamide: acetonitrile (1:2, $v/v$) leads to the formation of the CP [Zn(L1)(NMF)]$_n$ (**1**) [L1 = 5-{(pyren-4-ylmethyl)amino}isophthalate] (Scheme 2).

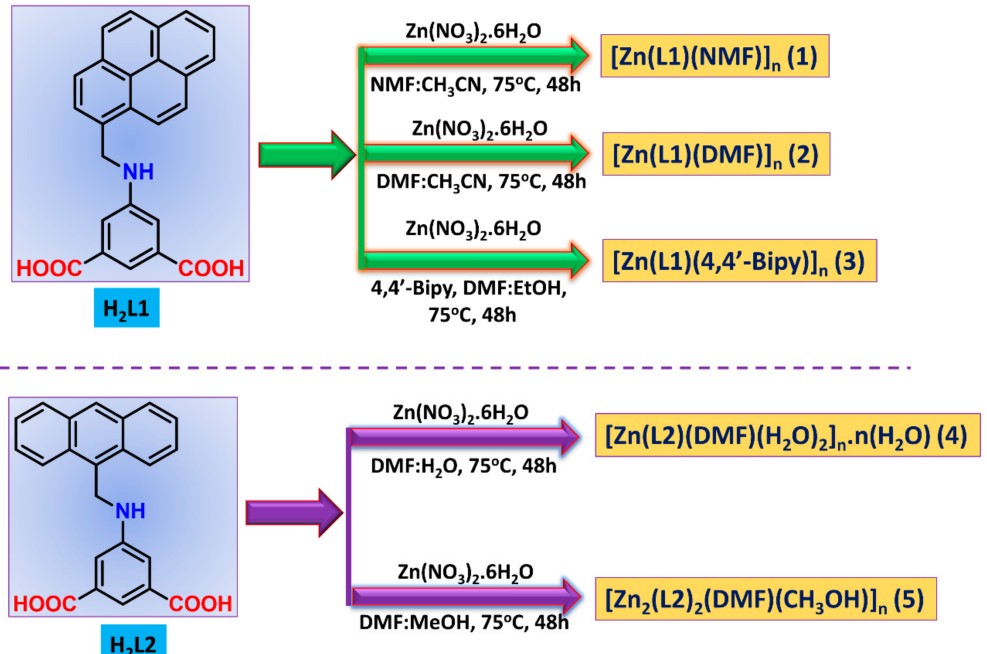

**Scheme 2.** Synthesis of coordination polymers **1**–**5** by reactions of $H_2L1$ and $H_2L2$ with Zn(NO$_3$)$_2$·6H$_2$O under various solvothermal conditions.

A mixture of **H$_2$L1** (10 mg, 0.025 mmol) and Zn(NO$_3$)$_2$·6H$_2$O (7.4 mg, 0.025 mmol) was placed in a 5 mL glass vessel, whereupon 2 mL of N-methylformamide:acetonitrile (1:2, *v/v*) mixture was added. This mixture was sealed and heated (hydrothermal reactor) at 75 °C for 48 h, after which it was cooled to room temperature giving colorless plate-like crystals of **1**. FT-IR (KBr, cm$^{-1}$) 3387 (m), 2826 (w), 1681 (s), 1598 (s), 1501 (m), 1423 (s), 1337 (m), 1304 (m), 1265 (s), 1133 (w), 1113 (w), 1076 (m), 941 (s), 849 (s), 755 (s), 781 (m), 711 (w), 667 (m), 483(w). Anal. Calcd. for C$_{27}$H$_{20}$N$_2$O$_5$Zn (M = 517.82): C, 62.62; H, 3.89; N, 5.41. Found: C, 62.23; H, 3.54; N, 5.34.

### 3.3. Synthesis of Coordination Polymers 2–5

The syntheses of coordination polymers **2–5** were carried out by the reported procedures [22]. The reactions of **H$_2$L1** with Zn(NO$_3$)$_2$·6H$_2$O in dimethyl formamide (DMF):CH$_3$CN or in DMF:EtOH, in the absence or presence of 4,4′-bipyridine under solvothermal conditions, led to the formation of [Zn(L1)(DMF)]$_n$ (**2**) and [Zn(L1)(4,4′-Bipy)]$_n$ (**3**), respectively (Scheme 2). The solvothermal reaction of **H$_2$L2** with Zn(NO$_3$)$_2$·6H$_2$O in a DMF:H$_2$O or a DMF:MeOH mixture led to [Zn(L2)(DMF)(H$_2$O)$_2$]$_n$·n(H$_2$O) (**4**) or [Zn$_2$(L2)$_2$(DMF)(CH$_3$OH)]$_n$ (**5**), respectively, [L2$^{2-}$ = 5-{(anthracen-9-ylmethyl)amino} isophthalate] (Scheme 2). The FT-IR, elemental, thermogravimetric and surface area analysis data are in agreement with those previously reported [22].

### 3.4. Procedure for the One-Pot Tandem Deacetalization–Knoevenagel Condensation Reactions

The ultrasonic experiments were undertaken in an ATU ultrasonic thermoregulated bath. Benzaldehyde dimethyl acetal (152 mg, 1.0 mmol) was added to a stirred mixture of malononitrile (132 mg, 2.0 mmol) and a catalytic amount of **1–5** (0.25–2 mol%) in dimethyl-formamide (0.5 mL). The reaction vessel was immersed in an ultrasonic bath, and the reaction mixture was allowed to stir at 50–80 °C for 0.25–2 h. Finally, 5 mL of CH$_2$Cl$_2$ was added, the catalyst was separated by centrifugation and, after evaporation of the filtrate, the crude product was isolated. The yields were determined by analysis of the $^1$H NMR spectra in CDCl$_3$ using mesitylene as an internal standard. An example of $^1$H-NMR spectra is presented in Figures S7–S11 (Supporting Information), and the reaction yield was calculated according to the internal standard method as reported in the literature [15]. In most cases, besides the final product, the benzaldehyde intermediate was also observed by NMR. We have used the atomic adsorption spectroscopy method for the eventual detection of the Zn(II) ion in the solution.

## 4. Conclusions

We have synthesized a new (CP **1**) and four other already reported (CPs **2–5**) Zn(II)-based coordination polymers from the reactions of two polyaromatic and amine groups containing carboxylic acid-based pro-ligands, namely 5-{(pyren-1-ylmethyl)amino}isophthalic acid (**H$_2$L1**) and 5-{(anthracen-9-ylmethyl)amino}isophthalic acid (**H$_2$L2**), with Zn(NO$_3$)$_2$·6H$_2$O, under solvothermal reaction conditions. Single crystal X-ray diffraction analyses revealed that CPs **1** and **2** are pseudo-polymorphic isomers and both 1D polymers but differ in the coordinating solvents (NMF for CP **1** and DMF for CP **2**). Framework **3** presents a three-dimensional network, the anthracene group containing CPs **4** and **5** have 1D and 2D networks, respectively.

The presence of Lewis acid (metal ion) and basic (amine group) centers, as well as the insolubility in organic solvents, make these CPs suitable for behaving as heterogenous bifunctional catalysts for the one-pot deacetalization–Knoevenagel condensation under ultrasound irradiation. They proved to be highly efficient, especially catalyst CP **1,** which completely converted the benzaldehyde dimethyl acetal into the desired product 2-benzylidenemalononitrile after 2 h, with a yield already of 80% after 0.5 h at 80 °C. Although not so effective, the remaining CP catalysts always led to yields above 90% for the same period of time. The highest catalytic activity of CP **1** is accounted for its low dimensional weak hydrogen-bonded structure and the lability of a coordinated NMF

molecule, favoring the accessibility and unsaturation of the metal centers. All the catalysts remained stable during the catalytic process and could be recycled at least up to five to six times with a slight loss of activity observed from the third cycle for low-dimensional 1D CPs (**1**, **2** and **4**) and from the fifth cycle for the higher-dimensional CPs (**3** and **5**). The current work shows that polyaromatic group bearing amine-functionalized CPs can perform as efficient heterogeneous bifunctional catalysts under mild conditions for the deacetalization–Knoevenagel cascade reactions. These results also afford a useful understanding of the applicability of CPs as important materials for the construction of new acid−base bifunctional heterogenous catalysts, which deserves to be explored in future.

**Supplementary Materials:** The following supporting information can be downloaded at: https://www.mdpi.com/article/10.3390/catal12030294/s1, Figure S1: Thermogravimetric analysis curve for CP **1**; Figure S2: One-dimensional structure of CP **2** (A) and three-dimensional structure of CP **3** (B); Figure S3: Crystal structures of one-dimensional CP **4** (A) and two-dimensional CP **5** (B); Figure S4: FT-IR spectra (A) and PXRD diffractograms (B) of catalyst CP **1** before and after the 4th recycling in tandem deacetalization–knoevenagel reactions; Figure S5: Yield of 2-benzylidenemalononitrile (**D**) upon catalyst recycling for the deacetalization-Knoevenagel reactions catalyzed by CPs **2**–**5**; Figure S6: PXRD diffractograms of catalyst CP **2** (A), CP **3** (B), CP **4** (C) and CP **5** (D) before and after the recycling in tandem deacetalization–knoevenagel reactions; Figure S7: $^1$H-NMR spectra of ultrasonic assisted one-pot tandem deacetalization–Knoevenagel reactions of benzaldehyde dimethyl acetal with catalyst CP **1** in CDCl$_3$ (entry 1, Table 1); Figure S8: $^1$H-NMR spectra of ultrasonic assisted one-pot tandem deacetalization–Knoevenagel reactions of benzaldehyde dimethyl acetal with catalyst CP **2** in CDCl$_3$ (entry 2, Table 1); Figure S9: $^1$H-NMR spectra of ultrasonic assisted one-pot tandem deacetalization–Knoevenagel reactions of benzaldehyde dimethyl acetal with catalyst CP **3** in CDCl$_3$ (entry 3, Table 1); Figure S10: $^1$H-NMR spectra of ultrasonic assisted one-pot tandem deacetalization– Knoevenagel reactions of benzaldehyde dimethyl acetal with catalyst CP **4** in CDCl$_3$ (entry 4, Table 1); Figure S11: $^1$H-NMR spectra of ultrasonic assisted one-pot tandem deacetalization–Knoevenagel reactions of benzaldehyde dimethyl acetal with catalyst CP **5** in CDCl$_3$ (entry 5, Table 1); Table S1: Crystal data and structure refinement details for CP 1; Table S2: Selected bond distances (Å) and angles (°) for compound **1**; Table S3: The optimization parameters for deacetalization–Knoevenagel tandem reactions between benzaldehyde and malononitrile with CPs **1**–**5** as catalysts and under ultrasound irradiation. References [29–32] are cited in the supplementary materials.

**Author Contributions:** Conceptualization, A.K.; investigation, A.K., M.M.A.S. and E.C.B.A.A.; methodology, A.K.; resources, E.C.B.A.A. and A.J.L.P.; writing—original draft, A.K. and E.C.B.A.A.; writing—review and editing, M.F.C.G.d.S. and A.J.L.P. All authors have read and agreed to the published version of the manuscript.

**Funding:** This work was supported by the Fundação para a Ciência e Tecnologia (FCT), Portugal, projects PTDC/QUI-QIN/29778/2017, UIDB/00100/2020 and LA/P/0056/2020 of Centro de Química Estrutural and Institute of Molecular Sciences as well as RUDN University Strategic Academic Leadership Program. The author A.K. is grateful to the FCT and IST, Portugal, for financial support through "DL/57/2017" (contract no. IST-ID/107/2018). The funding institutions to be acknowledged are indicated above (Funding).

**Data Availability Statement:** Not applicable.

**Acknowledgments:** This publication has been supported by the RUDN University Strategic Academic Leadership Program (recipient A.J.L.P., preparation). A.K. expresses his gratitude to Instituto Superior Técnico and FCT for Scientific Employment contract (Contract No: IST-ID/107/2018) under Decree-Law no. 57/2016, of 29 August.

**Conflicts of Interest:** The authors declare no conflict of interest.

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
