# Peer review of "Polyaromatic Carboxylate Ligands Based Zn(II) Coordination Polymers for Ultrasound-Assisted One-Pot Tandem Deacetalization–Knoevenagel Reactions"

_catalysts, doi:10.3390/catal12030294_

Round 1

Reviewer 1 Report

The authors present the preparation of one new coordination polymer of zinc and engage in a comparative study with four others the deacetalization-Knoevengel reaction. These are reasonably effective catalysts that are not the best in the literature but worth reporting.  I support publication pending the authors address one scientific concern and one concern about presentation.

My scientific concern is that I do not understand how the authors are calculating catalyst loading based on polymers with undetermined molecular masses. A clear explanation of how they know mol % for these coordination polymers is useful.   

My presentation concern is that this manuscript is very long for the content. I think few will actually read this, despite aspects being of interest to two distinct communities. In all sections except materials and methods, the volume of text could be reduced. For example, the abstract attempts to justify and contextualize the work to the point where the results are not even clear. Likewise, the comparative structural section (2.2) could primarily be condensed to a table.       

While it is not important for publication, I am curious as to why the authors did not obtain GPC, light scattering, or other data that may better indicate chain length.

Reviewer 2 Report

Dear Editor,

The author have attempted synthesis of Zn(II) based coordination polymers and its utilization in Ultrasound-assisted One-pot Tandem Deacetalization−Knoevenagel Reactions. I am ready to recommend the manuscript for publication after major revisions.

  • How leaching was determined? have you checked the splitting of catalysts?
  • Four cycles are not enough to claim the stability of catalysts.
  • How to suppose your catalytic system is superior to reported one
  • How your reaction is green when the catalysts are synthesized by solvothermal process explain the procedure instead of refence (22. Karmakar, A.; Paul, A.; Santos, I. R. M.; Santos, P. M. R.; Pantanetti Sabatini, E.; Gurbanov, A. V.; Guedes da Silva, M. F. C.; 476 Pombeiro, A. J. L. Highly Efficient Adsorptive Removal of Organic Dyes from Aqueous Solution Using Polyaromatic Group 477 Containing Zn(II)-based Coordination Polymers. Cryst. Growth Des. (submitted).)
  • Avoid self-citation.
  • FTIR spectra is not convincing (Figure S3 FT-IR spectra)
  • TGA results are contrary to Figure 1 needs a comprehensive justification.
  • In scheme 2 A—B step the H2O is confusing should be eliminated.
  • How much energy impart by ultrasound irradiation (50/60 Hz) and 80 °C in 2 hours. What will be the cumulative energy for this reaction?
  • Table for optimization of parameters should move to supporting information.
  • The manuscript should be reorganized all information regarding experiments like scheme 1 &2 and synthesis and catalysis detail must include in experimental section. The results need appropriate discussion, and the conclusion should write more precisely by removing repetition.  

Reviewer 3 Report

The paper reports the catalytic performance in deacetalization − Knoevenagel condensation for five Zn(II) metal-organic frameworks, including one new structure. The catalyst design idea is not very novel. However, the manuscript is very well written and important results are presented. Therefore, I can suggest only minor corrections to improve the article. 

1. All the primary NMR spectra, which were used for obtaining conversion and yield values, need to be added into ESI. 

2. Substitiution of NMF by DMF in 1 is possible during the contact with DMF-based solutions. What is the real composition of the catalyst? 

3. What about PXRD for 2-5 after several cycles? 

4. Please describe CHN and gas sorption instrumentation in the experimental. 

5. Which method was used for the detemination of 0.023% Zn content in leach liquor? This information should be added into the experimental part too. 

Round 2

Reviewer 2 Report

The manuscript has many spell/grammatic mistakes